# Volcanoes in video games: The portrayal of volcanoes in Commercial-Off-The-Shelf (COTS) video games and their learning potential.

Edward G. McGowan[1] and Jazmin P. Scarlett[2]

[1]School of Geography, Geology and the Environment, University of Leicester, Leicester, LE1 7RH, Leicestershire

[2]Formerly School of Geography, Politics and Sociology, Newcastle University, Newcastle Upon Tyne, NE1 7RU, Northumberland; now independent

*Correspondence to*: Edward McGowan (emcgowan1@hotmail.co.uk)

**Abstract**

Volcanoes are a very common staple in mainstream video games. Particularly within the action/adventure genres, entire missions (e.g. *Monster Hunter: Generation Ultimate,* 2018) or even full storylines (e.g. *Spyro: The Reignited Trilogy,* 2018) can require players to traverse an active volcano. With modern advancements in video game capabilities and graphics, many of these volcanic regions contain a lot of detail. Most video games nowadays have gameplay times in excess of 50 hours. *The Legend of Zelda: Breath of the Wild* (2017) for example brags a minimum of 60 hours to complete. Therefore, players can spend a substantial amount of time immersed within the detailed graphics, and unknowingly learn about volcanic traits while playing. If these details are factually accurate to what is observed in real world volcanic systems, then video games can prove to be a powerful learning tool. However, inaccurate representations could instil a false understanding in thousands of players worldwide. Therefore, it is important to assess the accuracies of volcanology portrayed in mainstream video games and consider whether they can have an educational impact on the general public playing such games. Or, whether these volcanic details are overlooked by players as they focus solely on the entertainment factor provided. We have therefore reviewed several popular commercial video games that contain volcanic aspects and evaluated how realistic said aspects are when compared to real-world examples. It was found that all the games reviewed had a combination of accurate and inaccurate volcanic features and each would vary from game to game. The visual aesthetics of these features are usually very realistic, including lava, ash-fall and lahars. However, the inaccuracies or lack of representation of hazards that come with such features, such as ash-related breathing problems or severe burns from contact with molten lava, could have great negative impacts on a player's understanding of these deadly events. With further investigations assessing the direct impact on the general public, there is the opportunity to correctly assess how to incorporate the use of mainstream video games in educational systems and outreach.

# 1. Introduction

## 1.1. Commercial Off-the-Shelf vs Educational Video Games

Video games can be categorised into different groups, based on playable design, graphic style or genre. The focus of this investigation will be on mainstream, or Commercial Off-the-Shelf (COTS) video games as opposed to educational games. Educational games have been intentionally designed to teach the player about particular topics. They are often developed with input from teachers to ensure the information included is factually correct, and sufficiently covers the topic of interest. While

the use of educational games has been heavily researched (e.g. Oblinger, 2004; Kerawalla and Crook, 2005; Squire, 2005; Van Eck, 2006; Squire et al., 2008; Charsky, 2010; Wiklund and Mozelius, 2013; Lelund, 2014; Chen, Yeh and Chang, 2015; Rath, 2015; Mozelius et al., 2017), most conclude that players, particularly children, tend to lose focus or enthusiasm to such games, nulling the educational benefits they could provide (Kerawalla and Crook, 2005; Van Eck, 2006; Charsky, 2010; Floyd and Portnow, 2012a, 2012b; Lelund, 2014). However, if designed and implemented appropriately, the opposite effect can happen

and improve user's learning (Mani et al., 2016). COTS on the other hand are designed primarily for entertainment and therefore, retain the focus of players for much longer (Squire, 2005; Van Eck, 2006; Squire, DeVane and Durga, 2008; Floyd and Portnow, 2012a; Turkay and Adinolf, 2012; Wiklund and Mozelius, 2013; Lelund, 2014; Mozelius, et al., 2017), with most modern COTS offering numerous hours of gameplay that can exceed 50 hours (e.g. *Legend of Zelda: Breath of the Wild; Witcher 3: Wild Hunt; The Elder Scrolls V: Skyrim*). COTS also have the advantage over educational games in their appeal

that enables them to reach millions more players around the world (Mayo, 2009; Floyd and Portnow, 2012b). While the benefits of learning through commercial video games may not be as obvious or as structured as standard learning through an educational system, when exposed to situations such as unplanned tests, students can surprise themselves with what they have learned from games such as improved knowledge of historical events after playing the *Assassin's Creed* series (Kline, 2020).

## 1.2. Using Educational Games to Teach STEM Subjects

The use of Educational video games is becoming an increasingly popular concept as a teaching method (Gros, 2007; Squire, 2008; Lelund, 2014). A key benefit of Educational games is that they are specifically designed with a tailored content that can be directly implemented within an educational setting. A level system that gets progressively more difficult as the student progresses within the game can also allow students to ease into a new situation as they build up their understanding of scientific concepts (Gros, 2015). Utilising a video games ability to change different controlling factors to generate differing outcomes

makes them a powerful tool for STEM subjects. Not only do the students gain hands on experience, but they can also gain immediate results, allowing them to explore how varying factors influence the outcome of experiments.

Shute et al (2013), created an Educational game called 'Newton's Playground'. The game required students to draw routes that allowed a green ball to reach a red balloon. Each of the methods that allowed the ball to advance further directly obeyed basic

rules of physics, including gravity and Newton's three laws of motion. Statistical analysis not only revealed that playing 'Newton's Playground' lead to an improved understanding of the physics concepts instilled in the game, but that the students who engaged more with the game, reaching the higher levels, showed the largest increase in post-test scores.

In another example, Pringle et al (2017) created a forensic science Educational video game that allowed university students to
conduct a full burial excavation, including doing background research, field reconnaissance and eventual excavating of potential sites. Feedback from the students stated that they found the game to be very useful, engaging and generally accepted to be an enjoyable experience. However, some students were concerned about using the game as a formal assessment because of peers who would struggle with the technology could suffer poor marks.

One of the major problems with using educational games lies within their development. In order to create a video game that has enough factual content to be properly implemented into a course and is engaging enough for the students, a considerable amount of time must be invested for the creating of, trailing and improving each game (Pringle et al, 2017). In addition to this, many educational games of this style are developed through funded research projects (Mani et al, 2016; Pringle et al, 2017) As a result, a considerable amount of funding would be required to mass produce educational games for widespread distribution
to schools/ universities.

### 1.3. Using COTS to Teach STEM Subjects

The major downside to COTS games is that because their focus is more on entertainment than education, they can contain numerous unrealistic, or inaccurate features, which could instil a false understanding of real-world dynamics within players. Such inaccuracies may be introduced into a game for a variety of reasons: cost and development times are too high, lack of
research conducted by the developers, or that it provides a higher entertainment value/risk factor than realistic expectations. However, with careful guidance, this issue could easily be overcome (Floyd and Portnow, 2012b).

Science and scientists themselves are not the most common staples in COTS games, usually showcased as singular characters that assign some objectives required to progress (e.g. the Academy Scholars in the *Monster Hunter* franchise) or the games are
developed as niche simulators (e.g. *Surgeon Simulator*). There has been some research in the representation of science, scientists and other types of people in COTS games however, for example the portrayal of technoscience (Dudo et al., 2014); biotechnology (Murdoch et al., 2011) and the representation of queer people of colour (Smith and Decker, 2016). However, this is not to say that realistic science does not exist within standard commercial games of other genres (e.g. adventure, shooter or racing games). Technological advancements in commercially designed games have allowed developers to simulate real-
world principles (Mohanty and Cantu, 2011). This makes games such as *Zoo Tycoon, Roller Coaster Tycoon, SIMS* and even

*Angry Birds* excellent candidates for improved learning of STEM related subjects, including mathematics, physics, chemistry and economics (Mayo, 2009; Sun, Ye and Wang, 2015; Klopfer and Thompson 2019).

Mohanty and Cantu (2011) used commercial PlayStation-3 games to teach physics to undergraduate students. Taking examples from games like *Shaun White Skateboarding*, students were asked to measure the speed of the character, and in *Little Big Planet*, students calculated the motion of projectiles launched from cannons they could build within the game. At the end of the study, comments from the participating students positively supported the notion of using video games as a teaching method. Many liked the ability to gain direct, first-hand experience of scientific concepts, and the study even led to tangential learning in one student who noted how inaccurate the physics mechanics of the main character in *God of War* were by breaking the first law of Newtonian mechanics.

Research by Gampell and Gaillard (2016) used a mixture of disaster education-oriented video games, *Stop Disasters*, *Disaster Watch*, *Inside Haiti* and *Earthquake Response* and two COTS games which have disaster elements, *Fallout* and *SimCity*, to see how they instil disaster awareness, portray hazards, vulnerabilities, capacities, disasters and disaster risk reduction. As well as game content, player motivation, skill building and social interactions within these games. Similar to the argument of this paper, findings suggest that video games have the potential to be positive learning tools to reinforce disaster risk reduction messages. A more recent study used constructivist learning theory to explore the ability of 'serious' disaster video games to create student participation in learning within schools, and findings state that teaching and learning processes for both teachers and students need to be considered more in terms of the pedagogic process for the ability of students to enable deeper discussions and engagement with the curriculum (Gampell et al., 2020).

The Science Hunters project (Hobbs et al., 2019) utilised the popular COTS game *Minecraft* to engage children in scientific subjects, including plant biology, animal adaptation, volcanology, flood management and much more. In the case of animal adaptations, children were tasked with creating an animal that had adapted to particular habitats, using the building blocks *Minecraft* provided. They would then have to explain their choices, such as using orange-coloured blocks to camouflage the creature in an orange-sand desert.

COTS cannot be used to completely replace standard teaching methods as they will not be structured or in-depth enough to cover a full syllabus. However, if correctly implemented to facilitate sessions as they have been done in the examples above, then the positives can be of great benefit to both students and staff involved (Van Eck, 2006; Floyd and Portnow, 2012b).

## 1.4. Geoscience Within Video Games

Despite the numerous investigations mentioned above, there have been very few specifically targeting geoscience-related learning via video games. Chen, et al. (2015) tested a self-designed Role Player Game (RPG), which was heavily focused on geoscience-themed research to help students with their curricular learning. From the results they found there to be no significant statistical difference in the scores between groups of students who played an RPG game compared to those who did not. However, as this was an educational game, opposed to a COTS game, the lack of knowledge gained by the students may be due to the style of the game not being entertaining and engaging enough (Chen, et al., 2015).

Another geoscience-related educational video game created was *St. Vincent's Volcano*, created by Mani et al., (2016). Developed as an educational game, *St. Vincent's Volcano* was intended to be used to enhance volcanic hazard education and communication to local communities around the real-world volcano La Soufrière, located in the Lesser Antilles. Candidates (both students and adults) took a quiz prior to playing *St. Vincent's Volcano* to establish a current understanding of local volcanic hazards. Afterwards they took part in a 6-week trial period playing the game before retaking the quiz. The results showed an increase of over 10% in the post-test results compared to pre-test results, as well as a genuine increase in the candidate's interest in volcanic hazards, both amongst the students and adults.

More recently, Hut et al. (2019) compared whether geoscientists or non-geoscientists had a greater ability to determine whether a landscape was real in a video game. The prompt behind the study was related to vast improvements in video game graphics, allowing for more wondrous natural environments (*Legend of Zelda: Breath of the Wild, Middle Earth: Shadow of Mordor, Red Dead Redemption*) and the amount of time players spend immersed in said games could pose as an opportunity for tangential learning (Hut et al., 2019). While geoscientists were able to correctly identify more images as being virtual or real than non-geoscientists, the results suggest that non-geoscientists are still capable of determining the difference to an extent that the potential of erroneous learning (the learning of wrong or false information in the belief that it is correct) is low. Therefore, this suggests there should be no risk in tangential learning of geological concepts even if incorrectly presented in a video game.

As briefly mentioned above, one of the major negative sides to using COTS as a form of tangential learning is that they can often contain inaccurate features that would misinform players and lead to erroneous learning (Rath, 2015; Mozelius et al., 2017; Hut et al., 2019). This could be due to a number of reasons, from the developers not fact-checking their sources, to the game being more entertaining when aspects are exaggerated. Previous research by Parham et al. (2010 & 2011) has already highlighted several volcano-topic areas where Hollywood films such as *The Core* have led to false understandings of our planet and volcanic eruptions in students. This includes belief that atmospheric changes can trigger volcanic eruptions and that said

volcanoes are only found in tropical environments (Parham et al., 2010). Therefore, it can be assumed that popular video games could also have a similar impact on student understandings of volcanic systems found within COTS games.

This paper is part one out of two, focusing on an overview of COTS video game educational potential. The second part of the investigation will be to explore what people do learn whilst playing COTS games. The aim of this investigation is to identify areas of volcanic features that are common, come across within COTS video games and apply real-world context to said features in order to determine how realistically they are presented. This will help to (1) highlight areas within volcano-related

teaching that players may pick up erroneous learning and (2) promote various COTS to increase their enthusiasm towards the subject and encourage further tangential learning (Floyd and Portnow, 2012b). The latter would not only have benefits within an academic teaching environment, but also in the use for outreach events.

## 1.5. Potential for Self-learning

Outside of the education system, video games have an amazing potential to stimulate self-learning. There are two particular

types of learning that can be induced from playing COTS. Tangential learning is the process of self-educating oneself through exposure to a topic in a context that they already enjoy (Floyd and Portnow, 2012a, 2012b). This can include a range of outputs from watching films and documentaries, to playing sports or games. With video games being so popular with millions of people around the world, the use of tangential learning by playing such games could prove to be a powerful tool for encouraging student interaction or boosting public engagement. *God of War* for example, has the potential to interest players in Greek

mythology as the players interact with various Greek deities and other mythological beings (Turkay and Adinolf, 2012).

Incidental learning refers to learning that is unplanned, often unconscious in nature, which develops through engaging in tasks or activities. In regard to video games players can become so focused on completing missions or drawn into the storylines that they do not register what they are learning at the same time. *Assassin's Creed 2* is set in Renaissance Italy, with several maps

that allow the player to fully explore the cities, learning about culture, politics and historical events of the time while progressing through the game's storyline (Turkay and Adinolf, 2012).

It is this potential of self-learning outside of educational environments, where the games are not forced upon the players, that has seen the least amount of analytical attention, and therefore the basis of exploration in this study. How much as players, do

we truly learn while casually playing a commercial video game for entertainment? And, because the information in these games are not fact-checked, how much of this information is scientifically accurate? For the purpose of this research, the investigation shall focus on volcanic systems and features found within the video games.

The hypothesis is that there will be a range of volcanic features represented. With volcanic regions being so prominent in COTS video games and volcanoes presenting a multitude of hazards in the real world, developers have access to a diverse pool to create unique environments and levels that will set them apart from other volcanic regions in other games. However, it is not expected that all of these volcanic features would be realistic, as COTS video games are designed with entertaining their audience in mind. They allow players to venture into virtual fantasy realms beyond the limits of our own. Therefore, many volcanic features found within the video games could be shaped into captivating landscapes or manipulated to provide a challenging, yet achievable task.

## 2. Method

To determine which volcanic features commonly occur in COTS video games, a variety (eleven in full and several partial reviews to date) of video games from popular franchises and titles were selected, including: *The Legend of Zelda, Pokémon, Spyro, Tomb Raider* and *Minecraft*. These games span an assortment of consoles, played on the Nintendo Switch, Xbox One and PC. Each game was chosen because they contain known extensive volcanic regions or levels that could be interacted with to make observations on the geological features found. Additionally, as with most COTS games, they have all been developed with player-entertainment in mind, opposed to primarily educating them.

Numerous hours were spent exploring the maps and levels that contain volcanoes looking at features including: texture, graphics and flow mechanics of lava (both molten and solidified); shape and eruption style of the volcano; hazard assessment/impact to local populations and more. With each example, comparisons were made against the visual representation in the video game to real-world examples. In cases where the game shows inaccurate representations, corrections were provided.

### 2.1. Selecting Video Games

Beyond the main requirement of the video games being popular COTS, we have made sure to include a broad range of game styles. Whilst most of the games we have chosen are part of the Role-Playing Games (RPG) and Action-Adventure genres, this is a fair representation of the current commercial video game market. However, video games also come in multiple forms: solo-player, multi-player, online, open-world, linear story etc. Each variant changes the gaming experience and as such would alter the way the player would learn. For example, in multiplayer games, a number of players will work together towards a common goal. Solo-player game play on the other hand will see an individual solve a problem on their own. It is therefore essential to cover such a range in order to understand how COTS can be best utilised for educational purposes.

## 3. Volcanic Features Within COTS

### 3.1. Volcano Shapes/ Styles

In the real-world, volcanoes can come in a range of shapes and styles (shield, stratovolcano, caldera, fissure etc). What is found
in video games is a preference towards stratovolcano or caldera styles (Table 1), and a definite lack of volcanic styles such as
shield or fissure. In most cases, stratovolcanoes present in-game are very large and steep-sided that tower over the landscape,
compared to real life counterparts, (Fig. 01). When the games require a volcano to cover a large area, they tend to opt for
calderas (*Legend of Zelda: Breath of the Wild* (2017) and *Subnautica* (2018)). As calderas are usually extensive, from 1 km to
40 x 75 km in diameter (Cole, et al., 2004), this is an understandable choice for developers. Whilst shorter volcanoes may not
look as dramatic as taller ones, they naturally produce larger scale lava flows/fields (e.g. Kīlaeau, Hawai'i or Laki, Iceland)
that are better attuned to the common video game representations of lava.

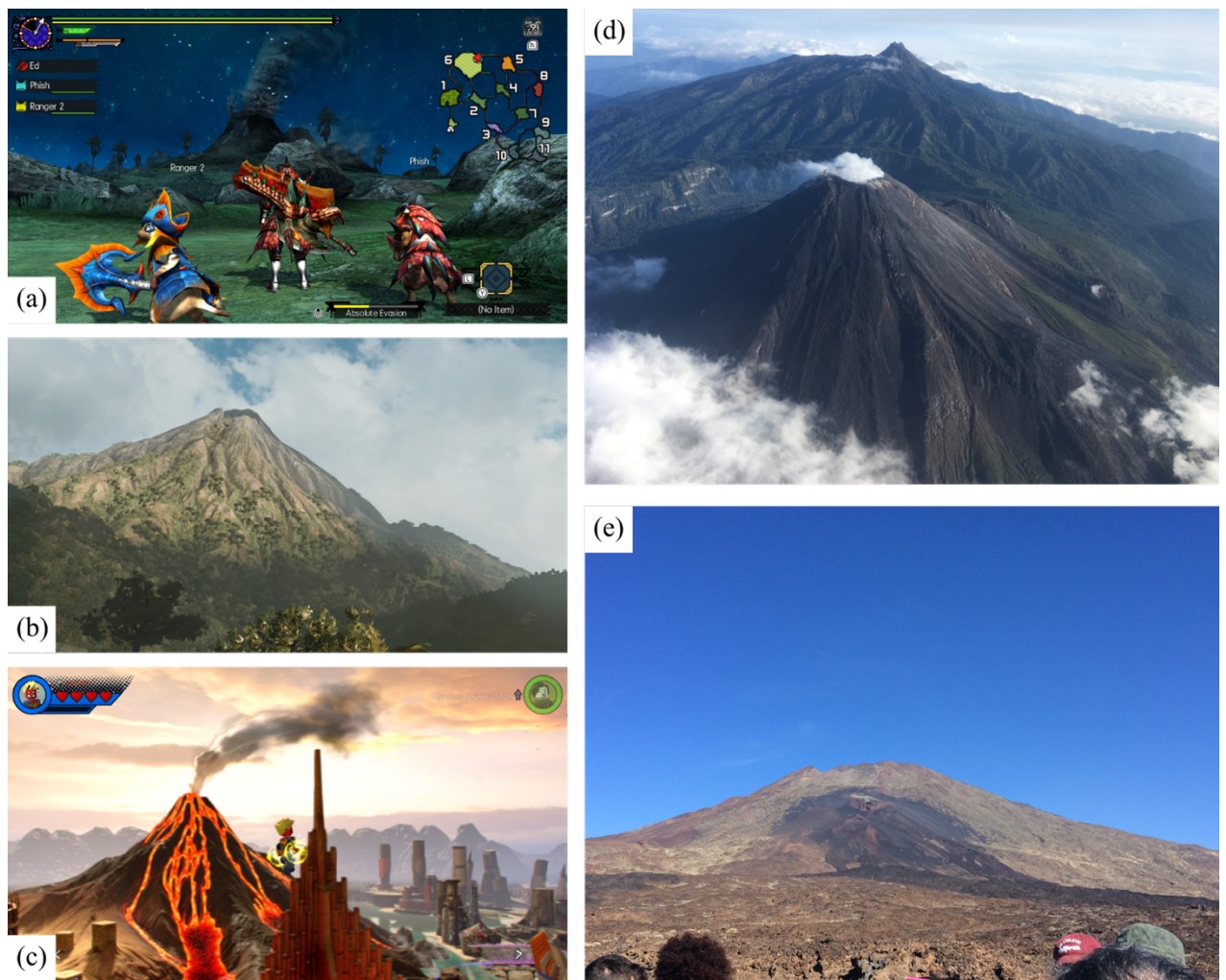

**Figure 01: Stratovolcanoes in *Monster Hunter: Generations Ultimat*e (2018; Fig 01a), *The Shadow of the Tomb Raider* (2018; Fig 01b) and *LEGO Marvel Superheroes 2* (2017; Fig 01c), aerial view of the stratovolcanoes Volcán de Colima (foreground) and Nevado de Colima (background) in Colima, Mexico (Edward McGowan, 2018; Fig 01d), view of active Las Cańadas summit from within Caldera de las Cańadas, Tenerife (Edward McGowan, 2016; Fig 01e).**

## 3.2. Lava flows

Lava flows are found to be the most represented volcanic feature within video games, appearing in nearly everyone reviewed (Table 1). Each depiction of lava flows was reasonably aesthetically accurate, including evidence of high viscosity and cooling surfaces (Fig. 02a). Even in the cases of solidified lava showing pillow lavas (Fig. 02c), ropey-like pāhoehoe (Fig. 02b and 02d), or columnar textures (Fig. 02e) could be found, adding to the different forms that players can learn about. The accuracy of the detail in each video game feature can be seen when compared to real-world examples of the same features (Fig 02f-h).

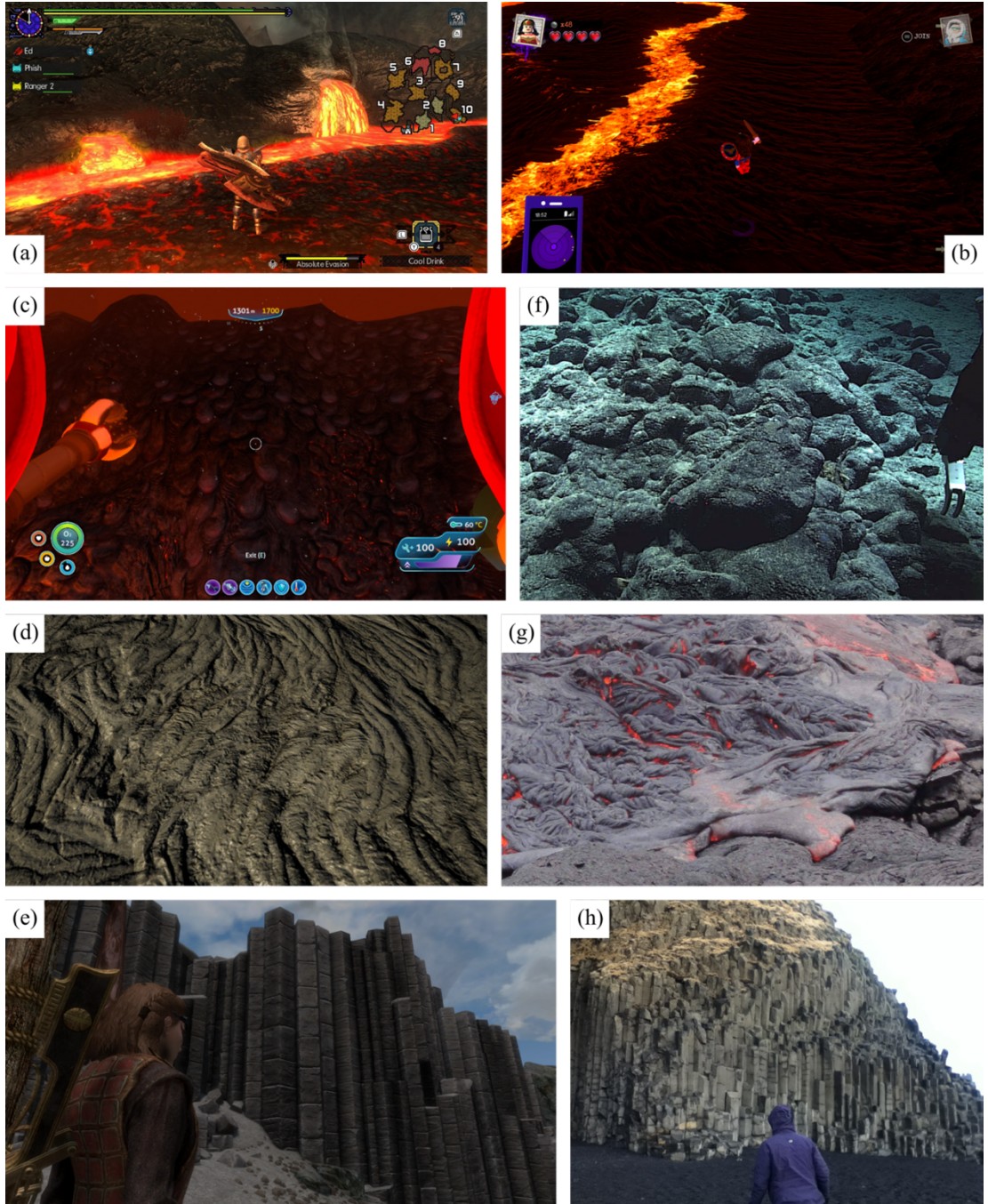

**Figure 02: Lava surfaces in *Monster Hunter: Generations Ultimate* (2018; Fig 02a), ropey-pāhoehoe textures in *LEGO DC Supervillains* (2018; Fig 02b), pillow lavas in *Subnautica* (2018; Fig 02c), more ropey- pahoehoe textures in *Assassin's Creed Odyssey* (2018; Fig 02d), columnar textures in *The Elder Scrolls: Skyrim* (2016; Fig 02e). Real-world examples of pillow lavas (Keoea Seamount, NOAA, 2015; Fig 02f), ropey- pahoehoe textures (Hawaii, Lis Gallant, 2013; Fig 02g) and basaltic columnar textures (Iceland, Edward McGowan, 2014; Fig 02h).**

Rivers of flowing lava or lava lakes on the other hand tend to be more exaggerated, either in their sheer size and length. *Legend of Zelda: Breath of the Wild* (2017) for example, boasts a two-tiered caldera, called Death Caldera/Mountain. Here the volcano has two caldera rims, both overflowing with lava flows to such a scale that is unfeasible in the real world (Fig. 03). Or in the case of *Subnautica* (2018), the lava was found to be flowing underwater at temperatures and colours found on the surface (Fig. 04), instead of quenching and forming pillow lavas, which is what it does in the real world (Fig. 02f).

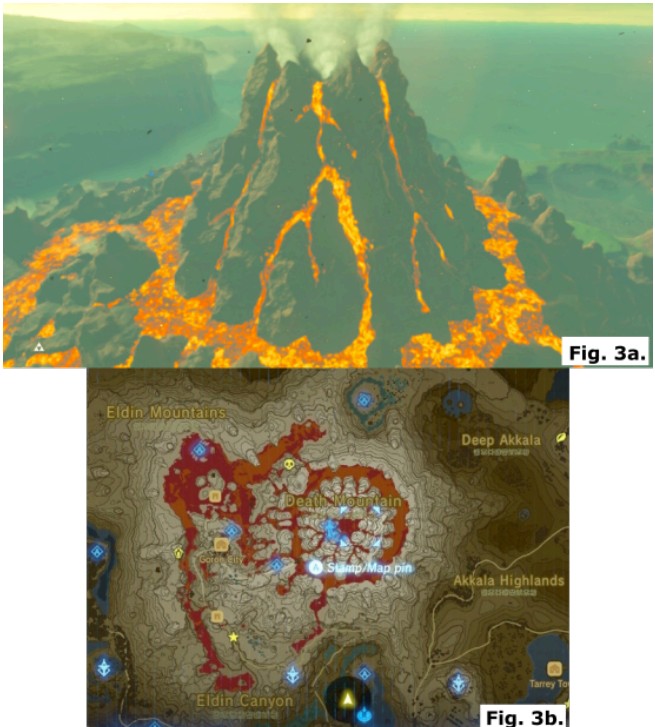

**Figure 03: The two-tiered caldera of Death Caldera/Mountain in *The Legend of Zelda: Breath of the Wild* (2017).**

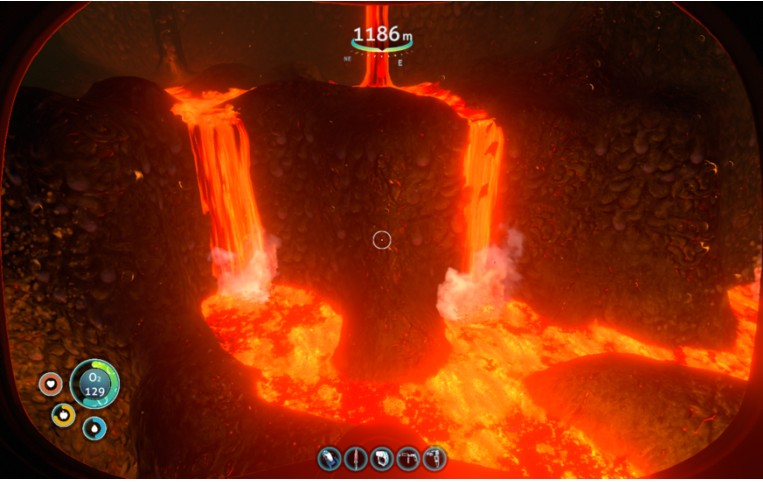

**Figure 04: Underwater lava flows in *Subnautica* (2018).**

## 3.3. Tephra

The most common representation of tephra in COTS video games is in the form of volcanic ash. While less synonymous with volcanoes than lava flows, volcanic ash is still a very common product of real-world volcanic eruptions (e.g. Eyjafjallajökull, Iceland, 2010 and Taal, Philippines, 2020). From a developer's point of view, volcanic ash is an easy volcanic aspect to edit into a game via a particle effect, as seen in *Pokémon Emerald* (2005)*, Legend of Zelda: Breath of the Wild* (2017) and *LEGO Marvel Superheroes 2* (2017; Fig. 05 and Video Supplement 01). In the real-world, volcanic ash can have detrimental effects on both social impacts (halting air-traffic, collapse roofs or destroying crops (USGS, 2019) and direct human health (respiratory problems and skin and eye irritation (Horwell and Baxter, 2006). Despite the common occurrence of volcanic ash in the games, these issues were rarely seen within the video games that were explored for this investigation. The best example found was within *Pokémon Emerald* (2005)*,* where the volcanic ash produced by the volcano Mt. Chimney was causing local residents to wheeze and cough due to breathing in volcanic ash for a prolonged amount of time (Fig. 06).

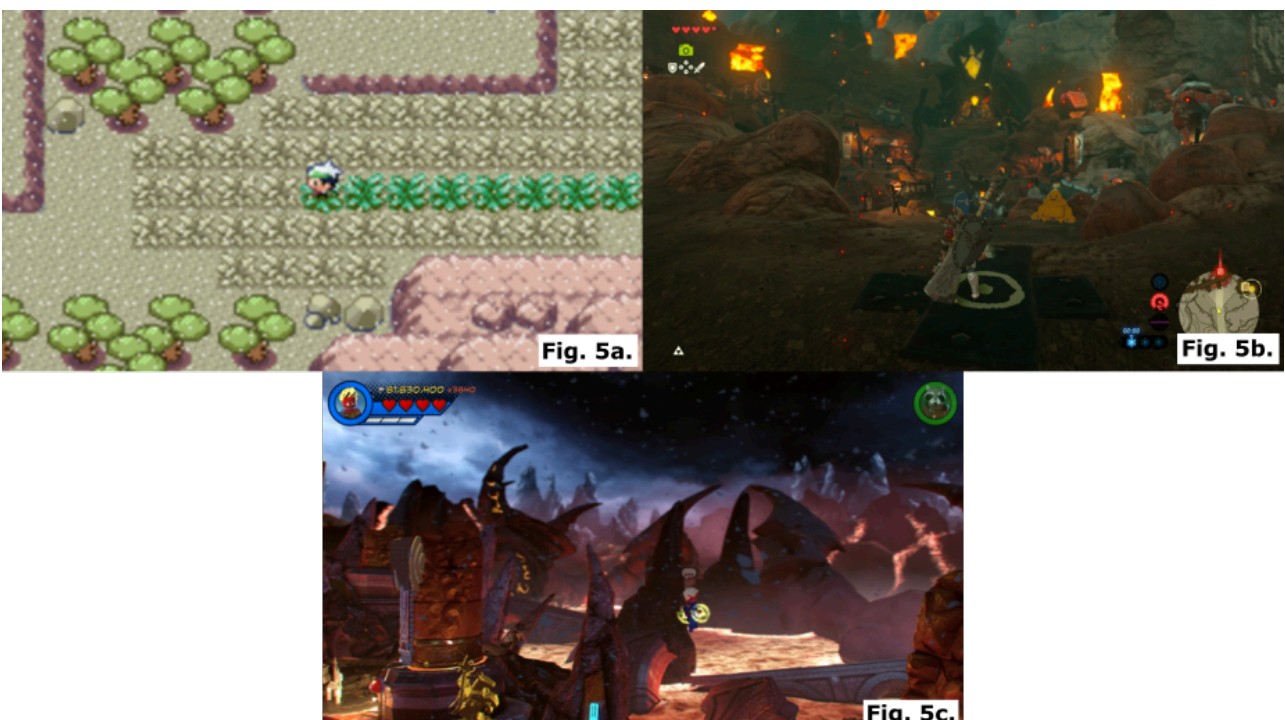

**Figure 05: Volcanic ash visual effects in** *Pokémon Emerald* **(2005; Fig. 05a),** *Legend of Zelda: Breath of the Wild* **(2017; Fig. 05b) and** *Lego Marvel Superheroes 2* **(2017; Fig. 05c).**

**Video Supplement 01: Ash particles in the air around Death Mountain in** *Legend of Zelda: Breath of the Wild* **(2017).**

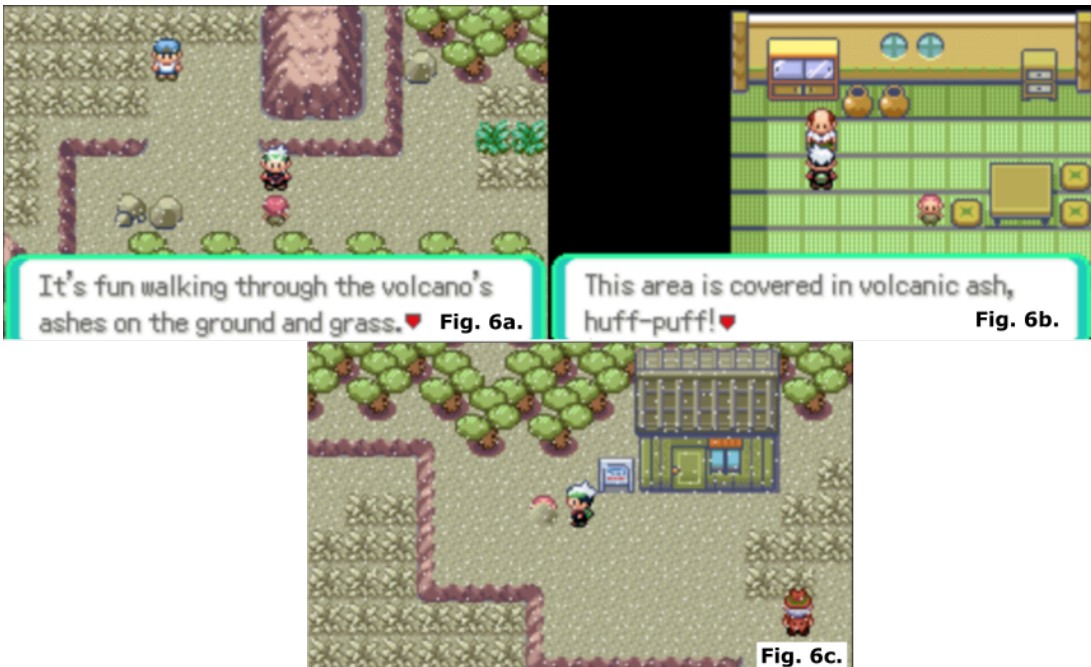

**Figure 06: The effects of volcanic ash from the volcano Mt. Chimney to local residents in *Pokémon Emerald* (2005).**

Lava bombs are another recurring volcanic feature, usually added as an additional hazard that players must avoid while traversing the stage (e.g. *Legend of Zelda: Twilight Princess* (2006)*, Spyro: Ignited Trilogy* (2018)*, Crash Bandicoot* (2018)). In the real-world, lava bombs are a serious threat to those within range. Some video games do take this risk seriously and directly apply the same level of severity (Table 2). If a player's avatar is hit by a lava bomb, they can instantly die, usually respawning them at the previous save point (Video Supplement 02, *Sea of Thieves* (2018)). However, in some games particularly aimed at a younger audience, the realism is reduced to make the level difficulty more appropriate for players. They do this by having the avatar only take some health damage, sometimes stumble backwards, but ultimately get back up again and continue on the path, dodging any further flying projectiles (*Legend of Zelda: Breath of the Wild* (2017)).

**Video Supplement 02: Player being hit by a lava bomb and instantly dying in *Sea of Thieves* (2018).**

Pyroclastic Density Currents (PDCs) are the least represented of the tephra-volcanic hazards within COTS video games. In the real-world PDCs are one of, if not the most, dangerous hazard a volcano can produce. They are also a phenomenal spectacle to watch as a cloud of molten rock and superheated gases avalanches down the slopes of a volcano. Despite the potential excitement and risk a PDC could provide in a video game, they are sorely lacking. The only hint of a PDC evident was in the artistic design of the cliffs surrounding some of the volcanic zones in *Monster Hunter: Generations Ultimate* (2018; Fig. 07a).

However, this is only an assumption made based on visual observations when compared to a real-world example (Fig. 07b).

There has yet to be seen a virtual PDC in motion.

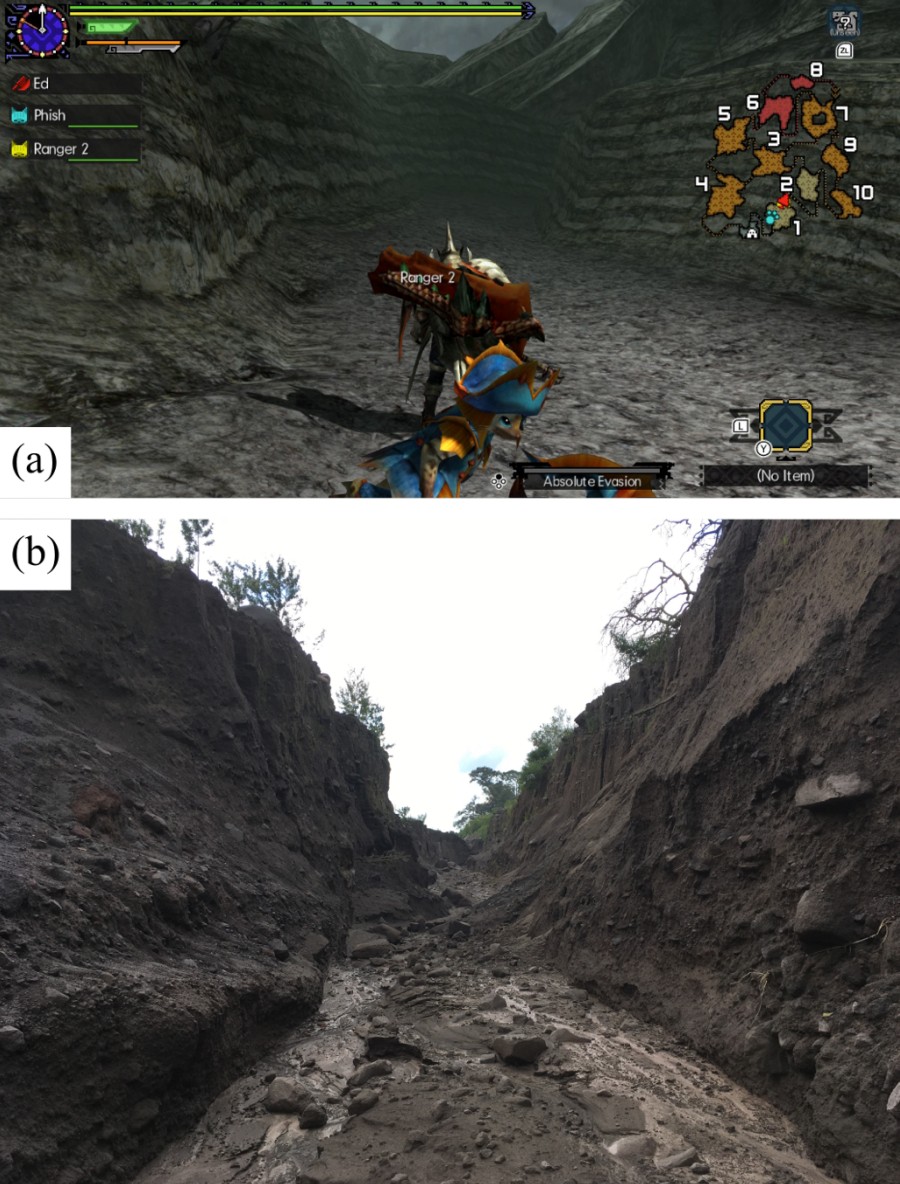

**Figure 07: Art design of potential PDC deposits in *Monster Hunter: Generations Ultimate* (2018; Fig 07a) compared to a real-world example of a dissected pyroclastic density current in Mexico, created by the Volcán de Colima eruption in 2015 (Edward McGowan, 2018; Fig. 07b).**

### 3.4. Lahars

Lahars are slurry mixtures of volcanic material, debris and water (or ice), that are highly erosive and damaging, can flow over gentle gradients and inundate areas far away from their sources, making them a distal volcanic hazard people sometimes do not anticipate (Wallace and Iverson, 2015). Whilst a common volcanic hazard for volcanoes that are ice/glacier capped (e.g. Nevado del Ruiz, Columbia and Mt. Rainier, USA), have a crater lake present (e.g. Taal Volcano, The Philippines and Ruapehu, New Zealand) or locations that experience heavy rainfall (e.g. Volcán de Colima, Mexico and La Soufrière, St.

Vincent and the Grenadines), lahars only feature in *The Shadow of the Tomb Raider* (2018; Fig. 08 and Video Supplement 03), being a sequence that must be traversed in order to progress in the game. The mechanics of the large flow itself were realistic, with the understanding of the dynamics between the ratio of sediment and water content and its bulldozing power by destroying property and infrastructure. However, the sudden opening of large cracks and gaps, subtracts from the realism in how lahars interact with the environment.


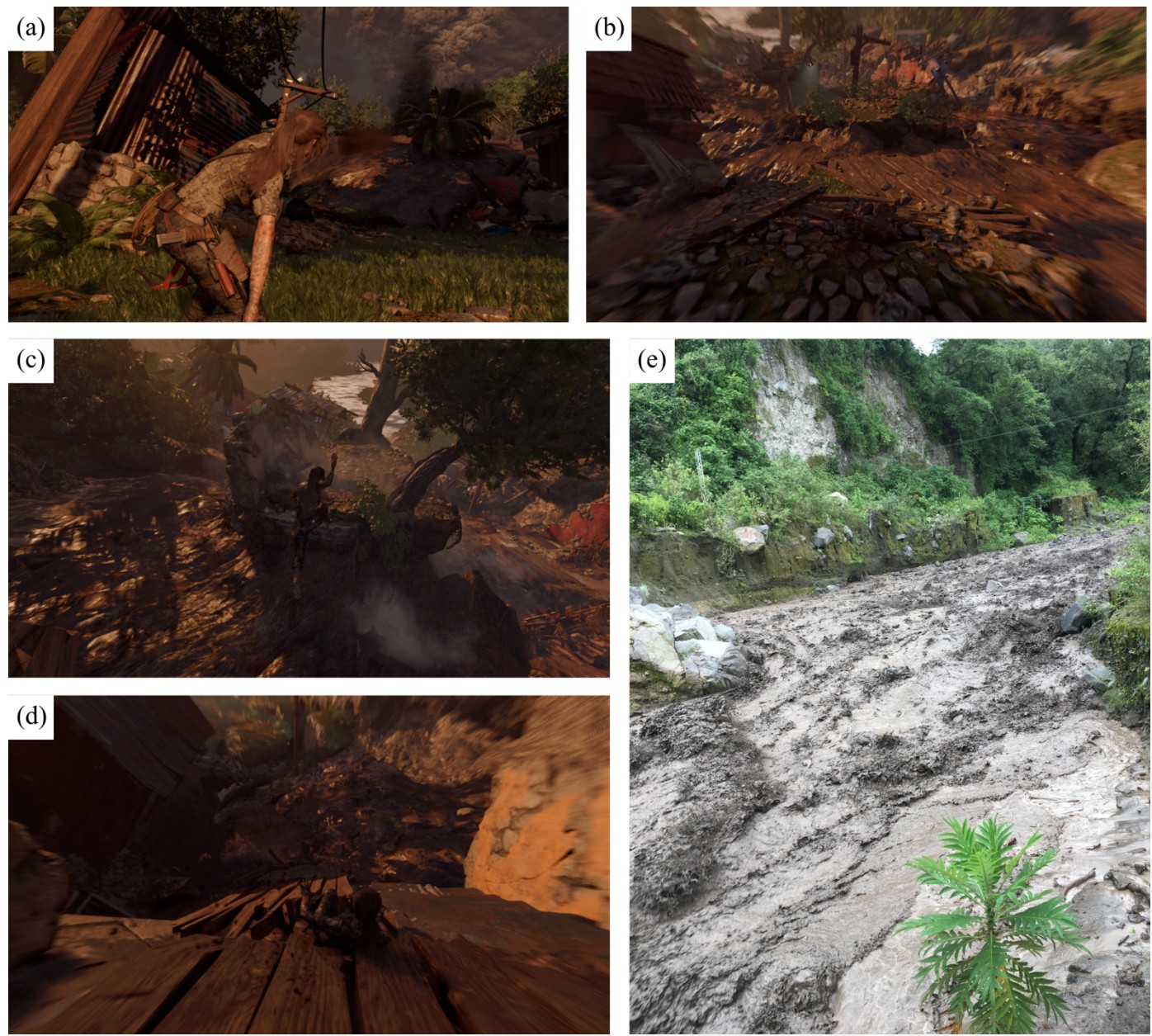

**Figure 08: Lahar sequence in _The Shadow of the Tomb Raider_ (2018; Fig. 08a-d) compared to a still image of a lahar in full flow down a barranca at the base of Volcán de Colima, Mexico (Edward McGowan, 2018; Fig. 08e).**

**Video Supplement 03: Two sequences of Lara (played character) trying to escape a lahar that rushes past her in _The Shadow of the Tomb Raider_ (2018).**

### 3.5. Volcanic Gas Emissions

Volcanic gases are by far the least represented aspect of volcanology, with barely a mention of them within the video games tested. However, in *The Shadow of the Tomb Raider* (2018), there is a sequence where the volcanic hazard is either volcanic
haze, tephra or a mixture of the two (Fig. 09), that results in the character covering their mouth, coughing and receiving slow damage and eventually death if lingering for too long. However, the confusion and perhaps misinterpretation of not being entirely clear if it is volcanic haze or tephra, could diminish tangential and/or incidental learning. This is disappointing because there is a very large portion of volcanological research being conducted on gas emissions to further our understanding of predicting eruptions, volcanic effects on climate and more. Without the inclusion of volcanic gases in COTS video games,
many players may never understand their importance.

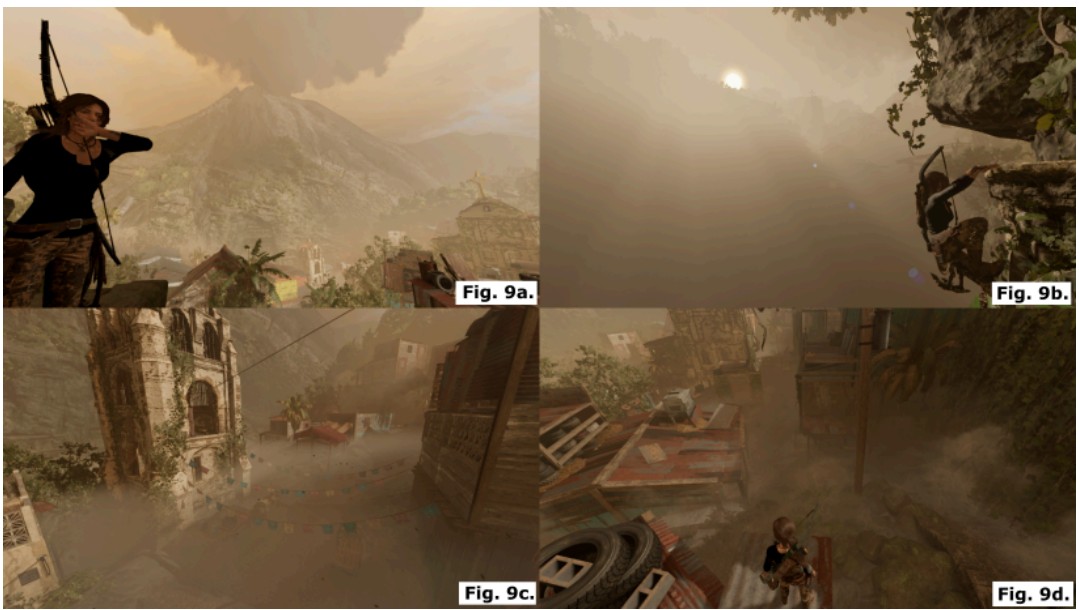

**Figure 09: Tephra and volcanic haze visual effects in *The Shadow of the Tomb Raider* (2018).**

## 4. Discussion

### 4.1. Overall Volcanic Representation

There is no doubt that volcanoes within video games provide entertaining, challenging and popular levels, appearing in
numerous respected franchise COTS games. Because of this, millions around the world will ultimately become immersed within these volcanic landscapes, rich in real-world features on a regular basis for a considerable number of hours.

What was found through playing such games is that there was no 'perfect' game in regard to the portrayal of volcanic features. Each game had a mixture of realistic and unrealistic features (Table 1 and 2; Fig. 10). In most cases of unrealistic features, they tend to be exaggerated, such as overly sized stratovolcanoes and large volumes of lava flows being constantly produced (Table 1). Whilst this does result in very stereotypical landscapes that all will be able to recognise as 'volcanic', it could also lead to a false belief that all volcanoes are of this shape and always have lava flows pouring out of them, which is far from the truth (Siebert et al., 2015). These exaggerations will be included to improve the entertainment and excitement factor of the gaming experience, ruling that to be more important than making them more realistic (Hut et al., 2019).

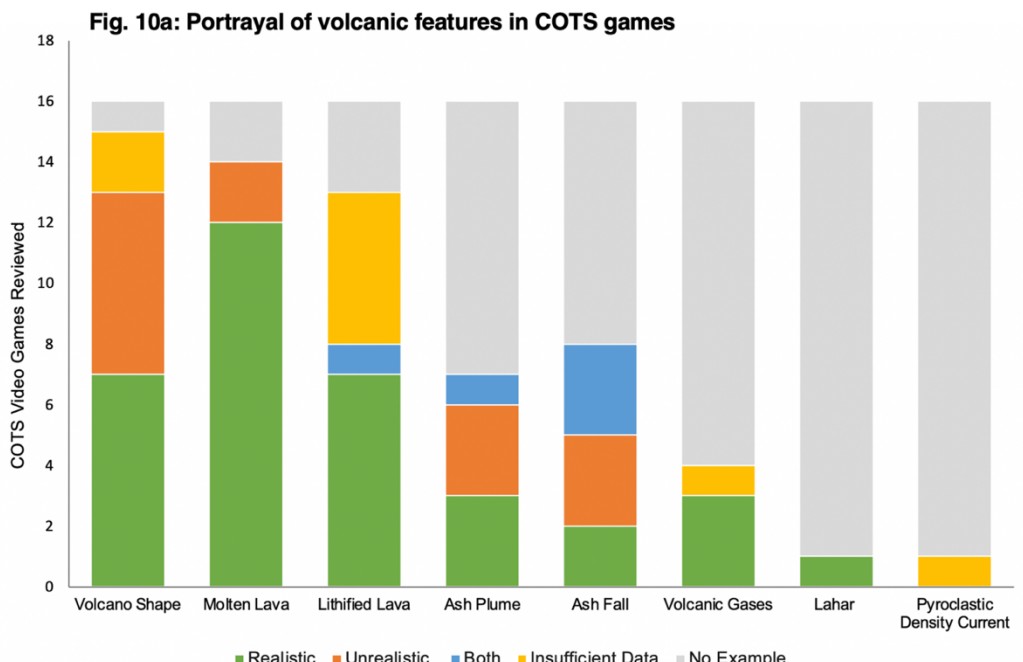

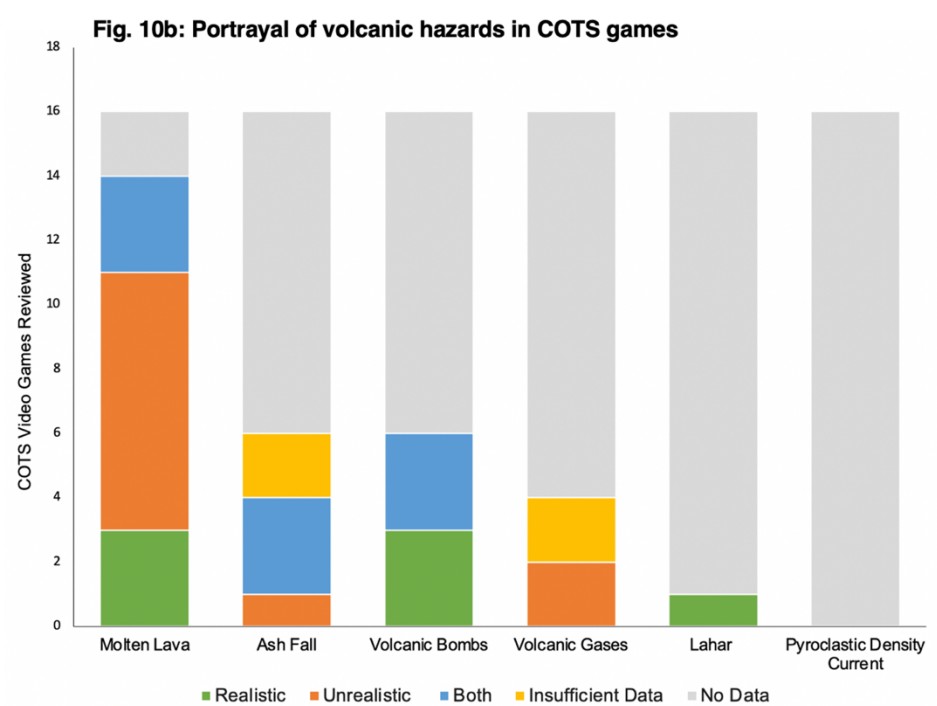

**Figure 10: The count of volcanic features (Fig. 10a) and of volcanic hazards (Fig. 10b) found within the COTS video games reviewed, displaying whether or not they were portrayed in a realistic or unrealistic manor.**

When it comes to the representation of volcanic features in the minds of the wider public, lava flows are synonymous with volcanoes. From the orangey-red glow to the natural power lava flows hold, it is always one to captivate an audience. There is also the danger that lava flows pose. Burning at hundreds of degrees Celsius, lava flows can destroy anything in its path, creating a great risk factor that developers can readily implement into their games for an added level of difficulty. Taking all of this into account, it is easy to see why they are such a staple in COTS video games (Fig. 10). However, it is the comparison between the appearances of volcanic features to those of volcanic hazards that highlights a problematic area (Fig. 10), not only in the number of realistic to unrealistic encounters but also in the number of encounters overall. In total, there are 78 representations of eight common volcanic features (Fig 10a), of these 35 are portrayed in a realistic manor. In contrast, there are only 31 representations of six common volcanic hazards (Fig 10b), and only 7 of them were realistically portrayed.

There are many potential reasons for developers focusing more on the visual aesthetics of a volcano than the hazards. One could be that the costs of implementing the damage mechanics is not seen as worthwhile. For example, PDCs are immense, catastrophic events that would require a great deal of animation development, potentially requiring a very dramatical moment of a story and even a cutscene to incorporate one. This would explain their low representation rate within the video games.

Another reason could be that as video game graphics are improving, especially in more modern video games, the developers want to focus on making the games more visually immersive, believing this is more important for players than their characters taking realistic damage.

The area that would initiate the least amount of tangential learning, despite their importance within volcanological geoscience, would be towards tephra fall deposits and gas emissions, in particular the risks both of these ejecta can cause. This is owed to the poor representation of the two topic areas within the COTS video games (Fig. 10), with volcanic ash often only presented as a particle effect and nothing more. As a result, players would be more likely to forget about the volcanic ash, ignoring it in a style similar to 'banner blindness' (Hervet et al., 2010), instead of gaining an interest in volcanic ash and prompting self-education into how it is formed and the major health risks it poses to life, or even the environmental benefits. Volcanic gas emissions are most likely under-represented due to their general colourless appearance, making them very difficult to visually implement within a video game environment. Therefore, it may be necessary to re-focus the educational curriculum if COTS games are to be used to include more on volcanic ash and volcanic gas emissions to ensure a well-rounded knowledge of volcanoes and to develop the ability for students to self-analyse their observations, such as asking them to hypothesis real-world risks within games (Van Eck, 2006; Parham et al., 2011).

## 4.2. Incorporating Learning Within Video Games

360 In-game tangential learning is an extremely effective way to utilise COTS video games. For example, *Subnautica* (2018) has an in-game encyclopaedia that registers information when travelling to certain biomes or obtaining materials. One such entry is about how the map is situated within the crater of an active caldera that collapsed thousands of years prior. With the use of in-game encyclopedias, players have the choice to access the information at any time, without having to stop playing at critical moments to actively research about the volcanic features they have just discovered. This not only provides easily accessible 365 information but does not hinder the entertainment factor of the gaming experience (Van Eck, 2006; Floyd and Portnow, 2012a).

Another method observed is the use of non-essential, non-playable characters (NPCs) that players do not need to interact with to progress, however, if placed near an interesting feature can provide further information about it when spoken to. Within 370 *Pokémon Emerald* (2005; Fig. 8a and Fig. 9) if players talk to NPCs on the slopes of Mt. Chimney or along Route 113, they will speak about the nearby active volcanic crater and the volcanic ash that it produces. However, it is vital that the information provided within the games are factually correct, otherwise players could take this information to be true (Parham et al., 2011). While the NPCs in *Pokémon Emerald* (2005) are very vocal about the falling ash, their lack of concern about ash-related health risks could easily lull players into a false understanding of real-world hazards.

375

Some video games have even made use of loading screens to add in quick facts while the player waits for the game to progress. These often cover game tactics and tips; however, they can also include information related to the setting of the game. For example, *Assassin's Creed: Odyssey* (2018) provides facts about Spartan history, Greek mythology, culture and major events that all tie into the time setting of the game (Brouwers, 2018). However, just like the potential use of NPCs, it is vital that these 380 quick facts are indeed factually correct. Otherwise, this would become another situation where commercial games would instil false understandings in players.

## 4.3. Impact of Incidental Learning

Incidental learning has a greater impact on erroneously learning with players unconsciously picking up details about volcanoes, such as the underwater flowing lava in *Subnautica* (2018) and without any direct in-game tangential learning mechanisms to 385 correct them, they could easily believe these errors to be fact. Repetitive erroneous incidental learning from non-traditional sources such as video games has statistically proven to lower the level of one's understanding about volcanic systems (Parham et al., 2011).

## 4.4. Utilising Video Games within Geoscience

A potential use in the education system could be to employ a style of facilitated learning. Differing from the other types of learning previously mentioned, facilitated learning encourages the students to take more control of their learning with the teachers providing resources and getting the students to discuss the situation themselves with minimal guidance. COTS would act as the resource, and the teachers could set assignments to students to play through similar video games and test their knowledge by getting them to assess the realism of the features found (Van Eck, 2006; Mohanty and Cantu, 2011), similar to what was done in this investigation. The idea of asking students to play a video game instead of reading a textbook would certainly prove to be more popular, and therefore potentially lead them to become more invested in their studies, while also improving their critical thinking skills (Parham et al., 2011; Hobbs et al., 2019). While COTS have the potential for driving tangential learning, providing students with a little guidance would provide even greater benefits by discussing and reassuring them that they are on the right path.

How different COTS video games are utilised within the education system would greatly depend on the age of the intended audience, their mentality to learning and their required curriculum. Examples of how COTS games can be utilised are as follows. Children-friendly games such as *Minecraft* (2009), which have access to multiplayer environments can make for the perfect games to get students within primary education interested in geoscience. The open-world setting would allow the young students to transverse a pre-made volcano in a group, letting them explore and show their peers different volcanic features that they find. The lessons would therefore be primarily student-led, with prompts from the teacher that direct students to key features when necessary. Maps on Minecraft (known as 'seeds') can also be shared to others, meaning the different volcanic or geological maps could easily be distributed to numerous institutions.

Within secondary schools, COTS can be used as homework tasks, where the student is asked to find an example of a geological feature within a COTS video game of their choice, and then have to write a short piece on their findings. Such tasks would allow the students to think outside the box with their learning and apply what they have been taught in the classroom to a very different setting. The open choice of video games means that students do not have to gain access to particular games and instead even use free to play games. Teachers as well would not have to make dramatic changes to their lessons plans as students would be playing the games at home, and yet allow said teachers to assess individual student's understanding of geoscience based on the work produced.

At a tertiary level (university/college) COTS can be applied to a wide range of scenarios as the style of education becomes more dependent of the students themselves. Multiplayer games can be used as a form of group-based projects, with each group being given a different game. The task would be for the students to work together to interpret geological features found within the games, interpret their formation, backed up with real-world examples they have researched and then compile their findings

into a group presentation. Solo-player games can be used for individual report-based coursework, acting in a similar way to the group-based project, student would instead write their findings up as a report to be marked by their assessor. This would be similar to the common 'field-based mapping' project that many students experience at university, even possibly acting as an additional alternative to the recently developed 'virtual field trips' by the University of Leeds (Houghton et al., 2015) and Imperial College London (Mackay, 2020). Such programs have been designed for students who are unable to go into the field for personal reasons (e.g. disabilities or health risks).

Public outreach is becoming increasingly important within the geoscience community, forming a core component of many geological societies, such as the Geological Society of London, the European Geological Union (EGU) and the American Geological Union (AGU) to name a few. However, while using traditional methods of publicly accessible peer-to-peer exchanges such as talks/presentations can prove effective, it is becoming evident that new, more modern methods may serve to enable wider public communication (Research Councils UK, 2008; Redfern, et al., 2016; Stewart and Lewis, 2017). This is where video games can be used to attract the public, using a medium they already understand and enjoy, and allow them to directly engage in geoscientific learning through playing. When attending events such as science fairs, members of the public can be given controls to individual games, allowing them to explore while geoscientists explain different features that are shown in a setting, they are more familiar with. Solo-player games with open-access worlds such as *Legend of Zelda: Breath of the Wild* (2017), would serve as a perfect example to use, as the public would be free to roam and the lack of objectives means that can come and go as they please without a feeling of pressure to complete the level. More COTS video games are also being developed for Virtual Reality (VR). If outreach events were to utilise such VR games, players could become full immersed within a geological setting, giving them a more hands-on experience than having to stand by and listen to complex geological information from a poster. The experience can even enable tangential learning as they can conduct follow-up research after the outreach event (Mohanty and Cantu, 2011; Mani et al., 2016). The greatest benefit of using COTS video games in this circumstance is that they have already been designed to be accessible and understandable for a large audience by communicating information with less technical jargon in an engaging manner, which is how geoscientific information to the public needs to be presented (Donnelly, 2008; Stewart and Lewis, 2017). In regard to volcanological geoscience, many do not have easy access to active volcanism, making COTS games an incredibly accessible way for people to interact with them in a relatively cheap (compared to costs involved in travelling to the site etc.), engaging and safe approach (Oblinger, 2004).

### 4.5. Considerations with Implementing Video Games within Geoscience Education.

The implementation of video game learning is not without its hurdles. People would need to be mindful that many COTS and the devices that operate them are expensive, and many might not have access to them. This is certainly a concern with more modern games that are becoming increasingly more expensive to purchase due to the higher developing costs to make the games more immersive. Considerations must also be taken into account in regard the age ratings assigned to each game. The

games we reviewed covered a range of age ratings, showing that there are games suitable for all, such as *Pokémon Emerald* (2002), which as an age rating of 3+ and so would make it suitable for primary education level teaching. However, *The Shadow of the Tomb Raider* (2018) has an age rating of 18+, meaning it would not be suitable for educational purposes to students until tertiary level.

There are also many cultural and social factors that influence gaming experiences, and therefore, the educational experiences as well. One example would be a player's familiarity with both gaming in general, and towards specific gaming titles or genres. For those who have more experience with playing COTS games, they will have an advantage with understanding the game mechanics, controls, level set ups and more. This advantage could make learning through video games more appealing to individuals. In contrast, those who are unfamiliar with gaming could find such tasks difficult and so become a less appealing method of learning to others (Pringle et al., 2017). Also, if the teachers are not familiar with playing video games, they may have to spend a number of hours familiarising themselves with game the play and testing the games to see if they are suitable for their teaching purposes before using them (Gros, 2015). As a result, video games could be seen as too much effort to implement to some educators.

Another consideration would be the ease of accessing particular areas within the video games. *Monster Hunter: Generations Ultimate* (2018) for example takes a considerable number of hours of gameplay in order to reach the levels that allow players to visit the volcano. Therefore, to save students from having to spend unnecessary time trying to reach particular areas, games would have to be chosen that either have direct access from the start, or game saves that have unlocked said areas would be required for easier implementation.

## 5. Conclusion

Commercial Off-the-Shelf video games contain a wide range of volcanic features, including lava flows, volcanic ash, lava bombs and even lahars, allowing millions around the world to interact with them in an entertaining environment outside of academia that could induce tangential learning.

As expected, these commercial games have a mixture of accurate and inaccurate features, with none showing to be flawless. Because of this, the use of COTS games for tangential learning should be done with caution. That is not to say it should be rejected entirely. Accurate features could be used within geoscience, particularly in regard to outreach work with the general public to capture an audience's attention without presenting misleading information, or to teach about volcanic hazards in a risk-free, engaging environment.

The inaccuracies within the video games tend to be over exaggerated in order to increase the entertainment factor, either by creating stunning landscape visuals or increasing the risk factor to provide more of a challenge for players. While this could lead to a lack of true understanding towards volcanic systems, advantages can be taken away from this. One would be to put a greater focus on volcanic hazards such as volcanic ash, volcanic gases and pyroclastic density currents, all areas that are wildly inaccurate at times or even non-existent, therefore likely to be forgotten or overlooked by students. The other would be to think in terms of tangential learning. The landscapes may be over-the-top at times, but they are also enticing to look at and admire. Through the enticement, players could be drawn into the appeal of volcanoes and take to other forms of learning platforms to teach themselves more about features they find within the games. Education systems could also take advantage of this and use the inaccuracies within the video games as tasked assignments through facilitated learning methods. Students could be asked to play through the games, find as many features as they can and comment on their realism.

While the extent of learning through playing COTS games is still unknown, the first part of our investigation has shown that these video games could indeed prove to be a useful source for future education to masses, both within academia and in outreach projects. The second part of the investigation on the learning potential of COTS games for volcanology will be to explore what people do learn. With further investigations assessing the direct impact on players, there is the opportunity to correctly assess how to incorporate the use of COTS games in geoscience.

**Author Contribution**

EM and JS conceptualized the project and developed the methodology together. EM carried out seven full game reviews, and JS carried out five full and four partial game reviews, with both validating the results together. JS prepared the figures for the manuscript. EM prepared the draft and editing of pre-publication manuscripts with contributions from JS.

**Competing Interests**

The authors declare that they have no conflict of interest.

Table 1: A summary of volcanic features in individual COTS video games. Features are colour-coded based on if they are realistic (green), unrealistic (orange), both realistic & unrealistic (blue), if there was insufficient data to determine accuracy (yellow), or no examples (white).

| Video Game | Volcanic Feature | | | | | | | |
|---|---|---|---|---|---|---|---|---|
| | Volcano Shape | Molten Lava | Lithified Lava | Ash Plume | Ash Fall | Volcanic Gases | Lahar | Pyroclastic Density Current |
| *Assassin's Creed: Odyssey* (2018) | Realistic composite/ stratovolcanoes | Realistic lava colours and textures with cooling surface | Realistic pahoehoe and ropey lava textures, matching the molten lava type | Convection of a plume and some atmospheric boundary spreading. However, no plume drift associated with wind interaction | No clear ash fall visual effects | N/A | N/A | N/A |
| *Crash Banicoot N. Sane Trilogy* (2018) | N/A | Realistic lava colours and textures with cooling surface | N/A | N/A | N/A | Possibly, there are small geothermal vents that make appearances | N/A | N/A |
| *Hot Lava* (2019) | No signs of a fissure vent to supply the lava | Realistic lava colours and textures with cooling surfaces | Realistic pahoehoe and ropey lava textures, matching the molten lava type | N/A | N/A | N/A | N/A | N/A |
| *LEGO DC Supervillians* (2018) | Cannot see crater to determine fully | Realistic lava colours and textures with cooling surface. | Realistic pahoehoe and ropey lava textures, matching the molten lava type | N/A | N/A | N/A | N/A | N/A |
| *LEGO Marvel Super Heroes 2* (2017) | Realistic composite/ stratovolcano | Realistic lava colours and textures with cooling surfaces in places | N/A | The top of the plume being directed away from vent due to prevailing winds | Constant falling of ash particles in the vicinity of the volcano | N/A | N/A | N/A |

| | | | | | | | | |
|---|---|---|---|---|---|---|---|---|
| ***Minecraft* (2009)** | Unrealistic lava pools randomly located around. No raised crater rim | Generic flowing lava colours and degassing bubbles. Decreases in volume as it flows further from source | Selection of several volcanic rocks including andesite and dacite, with realistic colours | N/A | N/A | N/A | N/A | N/A |
| ***Monster Hunter: Generations Ultimate* (2018)** | Unrealistically large strato-volcano. Mostly hollowed out on the inside | Unrealistically long, meandering lava river that does not match the expectations of the large volcano nearby | A range of lava textures that vary based on their location on the map. Although they have been smoothed over and lose detail | The top of several plumes show ash being directed away from vent due to prevailing winds | No ash fall is seen in any location, despite numerous nearby ash-laden plumes | N/A | N/A | Cliff faces closely resemble a dissected pyroclastic deposit |
| ***Pokémon Emerald* (2002)** | Realistic composite/ stratovolcano | Appears as generic lava colours and degassing bubbles | Appears as generic rock colours/texture | No ash plume seen above the active volcano, despite ash constantly falling | Constant falling of off-white coloured ash particles north of the volcano, suggesting a prevailing wind | Bubbling lava indicates expulsion of gases/ degassing | N/A | N/A |
| ***Pokémon Silver* (1999)** | Realistic composite/strato-island volcano | Appears as generic lava colours and degassing bubbles | Appears as generic rock colours/texture | N/A | N/A | Bubbling lava indicates expulsion of gases/ degassing | N/A | N/A |
| ***Sea of Thieves* (2018)** | Realistic composite/strato-island volcano | N/A | N/A | N/A | N/A | Gases escape fissures when the volcano erupts | N/A | N/A |
| ***Spyro: The Reignited Trilogy* (2018)** | Cannot see crater/source to determine fully | Appears as generic lava colours and degassing bubbles | Appears as generic rock colours/texture | N/A | Some particles are ash, most are cinders/embers | N/A | N/A | N/A |

| | | | | | | | | |
|---|---|---|---|---|---|---|---|---|
| *Subnautica* **(2018)** | Realistic collapsed caldera spanning 2 km in diameter | Molten lava flowing underwater with no cooling effects | Well detailed pillow lava and pahoehoe textures | N/A | N/A | N/A | N/A | N/A |
| *The Elder Scrolls: Skyrim* **(2016)** | Unrealistically steep cone-shaped | N/A | Detailed basalt columns | Lacks convection complexity and detail. Small amount of drift available | No ash fall however, ash deposits are light grey coloured | N/A | N/A | N/A |
| *The Legend of Zelda: Twilight Princess* **(2006)** | Unrealistically steep cone-shaped | Realistic lava colours and textures with some cooling surfaces around the edges | Appears as generic rock colours/texture | N/A | N/A | N/A | N/A | N/A |
| *The Legend of Zelda: Breath of the Wild* **(2017)** | Unrealistically large, two-tiered caldera. The central vent is also ridiculously steep | Realistic lava colours and textures with some cooling surfaces in low velocity areas | Appears mostly as generic rock texture. However, some areas show dark, cooled lava flows | Generic dark, pixelated cloud radiating out in all directions a short distance from the central vent | Ash depicted as a ember-like particles rising in the air | N/A | N/A | N/A |
| *The Shadow of the Tomb Raider* **(2018)** | Realistic composite/strato island volcano | Realistic lava colours and textures with some cooling surfaces around the edges | Appears as generic rock colours/texture | Show major convection, one has indication of drifting due to wind direction | First volcano example has more light showering of ember-like particles. Second example is more fog effect | N/A | Realistic flow showing material to water ration rheology and power | N/A |

**Table 2: A summary of volcanic hazards in individual COTS video games. Features are colour-coded based on if they are realistic (green), unrealistic (orange), both realistic & unrealistic (blue), if there was insufficient data to determine accuracy (yellow), or no examples (white).**

| Video Game | Volcanic Hazards | | | | | |
|---|---|---|---|---|---|---|
| | Molten Lava | Ash Fall | Volcanic Bombs | Volcanic Gases | Lahar | Pyroclastic Density Current |
| *Assassin's Creed: Odyssey* **(2018)** | Players can stand on the lava and take fire damage before dying | N/A | N/A | N/A | N/A | N/A |
| *Crash Banicoot N. Sane Trilogy* **(2018)** | Players take burn damage three times and then disintegrate | N/A | N/A | Small geothermal vents are present and take burn damage similar to the molten lava | N/A | N/A |
| *Hot Lava* **(2019)** | Players slowly sink into the viscous lava and vision whites out before level is reset | N/A | N/A | N/A | N/A | N/A |
| *LEGO DC Supervillians* **(2018)** | Depending on the character, take damage until being destroyed | N/A | N/A | N/A | N/A | N/A |
| *LEGO Marvel Super Heroes 2* **(2017)** | Depending on the character, take damage until being destroyed | No mention, or concern expressed towards the constant falling ash | Depending on the character, the bombs are one-hit kills | N/A | N/A | N/A |
| *Minecraft* **(2009)** | Quick over-time damage dealt. High viscosity makes escaping difficult. Fire-damage still dealt afterwards | N/A | N/A | N/A | N/A | N/A |

| | | | | | | |
|---|---|---|---|---|---|---|
| *Monster Hunter: Generations Ultimate* **(2018)** | Burn-damage dealt when standing too close to molten lava | Ash is blown away from accessible areas | N/A | N/A | N/A | N/A |
| *Pokémon Emerald* **(2002)** | Numerous people standing on the flanks of an active volcano, all the way up to the summit, in hopes of seeing it erupt | Some locals experience breathing difficulties. Vegetation covered in ash fall. Very limited concern with locals about the hazards the ash presents, as even children play in ash piles | N/A | No mention, or concern towards the volcanic gases when lava is clearly degassing next to people | N/A | N/A |
| *Pokémon Silver* **(1999)** | Pushing boulders into lava from a height within a public building causes no concern or harm to people standing within the splash-zone | N/A | Volcanic bombs block off access along a route. Locals are shown to be concerned. However, the bombs are located a considerable distance from the volcano | No mention, or concern towards the volcanic gases when lava is clearly degassing next to people, within a building that shows no ventilation systems | N/A | N/A |
| *Sea of Thieves* **(2018)** | N/A | N/A | Large bombs if on target, are one-hit kills. If off target, take damage | N/A | N/A | N/A |
| *Spyro: The Reignited Trilogy* **(2018)** | Player takes burn damage three times, turns black from burns and then slowly sinks into the lava | N/A | Non-hazardous to the player, but can be used as a weapon to one-kit kill enemies | N/A | N/A | N/A |

| | | | | | | |
|---|---|---|---|---|---|---|
| **Subnautica (2018)** | Low damage over-time dealt, even when swimming through the lava. Also, no change in viscosity when moving from water to lava and back, so players can easily escape | N/A | N/A | N/A | N/A | N/A |
| **The Elder Scrolls: Skyrim (2016)** | N/A | N/A | N/A | N/A | N/A | N/A |
| **The Legend of Zelda: Twilight Princess (2006)** | Player sinks into the lava very quickly. However, this is reasonable given the heavy inventory carried. Wooden equipment is destroyed. Relatively low damage dealt after respawn | Ash fall appears to be of no concern to local non-humans. However, this could be an evolutionary trait. Humans avoid the area until eruption subsides | Being hit by a lava bomb does reasonable health damage, scaled to the size of the bomb. Largest bombs can one-hit kill | N/A | N/A | N/A |
| **The Legend of Zelda: Breath of the Wild (2017)** | Player sinks into the lava very quickly. However, this is reasonable given the heavy inventory carried. Relatively low damage dealt after respawn | Ash fall appears to be of no concern to local non-humans. However, this could be an evolutionary trait. Humans avoid the area until eruption subsides | Being hit by a lava bomb does substantial health damage. Also, locals fear the economic damage of the eruption as tourists are too scared of the lava bombs to visit | N/A | N/A | N/A |
| **The Shadow of the Tomb Raider (2018)** | Player 'disappears' and instantly dies | A possible mixture of ash and gases cause slow breathing damage before dying. Inconclusive as cannot determine the cause | N/A | A possible mixture of ash and gases cause slow breathing damage before dying. Inconclusive as cannot determine the cause | Player is insta-killed from the debris within the flow and swept away | N/A |

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
