# Peer review of "Volcanoes in video games: The portrayal of volcanoes in Commercial-Off-The-Shelf (COTS) video games and their learning potential."

_Geoscience Communication, 2020_

## Referee Comment (RC1) · Ian Turner (Referee) · 11 Oct 2020

Dear Authors, I welcome your paper that explore the educational potential of commercial off-the shelf (COTS) as a learning resource. It was interesting to read, offered new insights, covered an area I have not previously read about in the literature and most importantly got me thinking. I look forward to the second paper you mention in the introduction. The methodology is a simplistic evaluation but acts as a suitable vehicle for the papers discursive nature.
Some questions that I feel that may enhance the papers overall aim of exploring the general potential of COTS for geoscience education.

1. You mention that COTS by their nature include some barriers to accessibility such as game cost. There are other in the same theme such as the cost of the console / device to run the game. Other barriers include age, as some of the games reviewed in this article have specific age guidance which could preclude them being used in specific educational settings. The cultural and social factors that influence gaming (perceptions of, use of etc.) are vast and beyond the scope of this article. However, they could be acknowledge as being important in the educational value of COTS video games. In its simplistic form familiarity with COTS or even the specific titles used in any educational initiative create 'capital' that alters the experience.

2. I feel that game play habits /dynamic offer a range of different perspectives on the potential of COTS videogames in this context. Game are typically solo or multiplayer often with the presence of an online community as seen in MMORPG. Many of the commercially successful COTS video games of recent years have a strong online community that offers new opportunities for geoscience education. In the latter for example climate change scientists have used the game Fortnite to talk about climate issues with players whilst inclined in the game. In educational settings group and team play offers a different set of opportunities to solo play.

3. The authors expertise on volcanoes allows them to comment on the accuracy of the content of the games. A novice player is unlikely to have the capital to make these judgements. Therefore, consideration of not just the 'could' but the 'how' COTS can be utilised in education to provide the critical skills needed to make the judgements would be welcome.

4. The paper uses a range of examples of educational impact – I think further differentiation of the potential aligned to educational levels (primary, secondary, tertiary) and outreach activity may help clarify where the largest potential impacts lie. Interactive comment

---

## Referee Comment (RC2) · Jamie Pringle (Referee) · 12 Oct 2020

Review of McGowan & Scarlett video game paper:

Firstly I would commend the authors on the paper, certainly current public, especially those under 30, will have grown up with computer gaming, and thus will be already familiar with the learning environment, and thus will have the capacity to learn within it.

I have made specific comments on an annotated PDF, general ones are below:

[Figure]

1) The introduction gives plenty of knowledge of relevant games, Im wondering whether you are missing a section on other educational egames in STEM? A couple here for you for info but there will be others: Pringle, J.K., Bracegirdle, L. & Potter, J. 2017. Educational forensic e-gaming as effective learning environments for HE students. In: Williams, A., Cassella, J.P. & Maskell, P.D. (eds.); Forensic Science Education & Training, Wiley Press. ISBN: 978-1-118-68923-3. Pringle, J.K. 2014. Educational egaming: the future for geoscience virtual learners? Geology Today, 30(4), 145-148. http://doi.org/10.1111/gto.12058 2) Are all of these games solo linear story ones? Or multi-threaded or even open world ones? Im unsure as not familiar with all the games. As peer-to-peer learning is surely important as well with group-based games as other pedagogic articles point out. 3) You have gone through the main aspects of volcano hazards well, with illustrations (and I like the supplementary video, would it be too much work to have more supplemental videos of all the key volcanic hazards do you think? Particularly as all the games are all 3D.. 4) Im also wondering if a summary table matrix would be of use to include in your paper, summarising (in sequential columns) the different key volcanic hazards, games which cover them well, and then which cover them badly? And anything else you deem relevant? That would be of real use to the reader and bring together your key findings of this paper I think? I hope the other reviewer is a volcanologist by the way who may note a key one which you have missed as Im not!

Hope helpful, Best regards, Jamie.

[Figure]

**Volcanoes in video games: The portrayal of volcanoes in Commercial-Off-The-Shelf (COTS) video games and their learning potential**

Edward G. McGowan[1] and Jazmin P. Scarlett[2]

[1]School of Geography, Geology and the Environment, University of Leicester, Leicester, LE1 7RH, Leicestershire
[2]Formerly School of Geography, Politics and Sociology, Newcastle University, Newcastle Upon Tyne, NE1 7RU, Northumberland; now independent

*Correspondence to*: Edward McGowan (emcgowan1@hotmail.co.uk)

**Abstract**

Volcanoes are a very common staple in mainstream video games. Particularly within the action/adventure genres, entire missions (e.g. *Monster Hunter: Generation Ultimate*, 2018) or even full storylines (e.g. *Spyro: The Reignited Trilogy*, 2018) can require players to traverse an active volcano. With modern advancements in video game capabilities and graphics, many of these volcanic regions contain a lot of detail. Most video games nowadays have gameplay times in excess of 50 hours. *The Legend of Zelda: Breath of the Wild* (2017) for example brags a minimum of 60 hours to complete. Therefore, players can spend a substantial amount of time immersed within the detailed graphics, and unknowingly learn about volcanic traits while playing. If these details are factually accurate to what is observed in real world volcanic systems, then video games can prove to be a powerful learning tool. However, inaccurate representations could instil a false understanding in thousands of players worldwide. Therefore, it is important to assess the accuracies of volcanology portrayed in mainstream video games and consider whether they can have an educational impact on the general public playing such games. Or, whether these volcanic details are overlooked by players as they focus solely on the entertainment factor provided. We have therefore reviewed several popular commercial video games that contain volcanic aspects and evaluated how realistic said aspects are when compared to real-world examples. It was found that all the games reviewed had a combination of accurate and inaccurate volcanic features and each would vary from game to game. The visual aesthetics of these features are usually very realistic, including lava, ash-fall and lahars. However, the inaccuracies or lack of representation of hazards that come with such features, such as ash-related breathing problems or severe burns from contact with molten lava, could have great negative impacts on a player's understanding of these deadly events. With further investigations assessing the direct impact on the general public, there is the opportunity to correctly assess how to incorporate the use of mainstream video games in educational systems and outreach.

**1. Introduction**

**1.1. Commercial Off-the-Shelf vs Educational Video Games**

Video games can be categorised into different groups, based on playable design, graphic style or genre. The focus of this investigation will be on mainstream, or Commercial Off-the-Shelf (COTS) video games as opposed to educational games. Educational games have been intentionally designed to teach the player about particular topics. They are often developed with input from teachers to ensure the information included is factually correct, and sufficiently covers the topic of interest. While the use of educational games has been heavily researched (e.g. Oblinger, 2004; Kerawalla and Crook, 2005; Squire, 2005; Van Eck, 2006; Squire et al., 2008; Charsky, 2010; Wiklund and Mozelius, 2013; Lelund, 2014; Chen, Yeh and Chang, 2015; Rath, 2015; Mozelius et al., 2017), most conclude that players, particularly children, tend to lose focus or enthusiasm to such games, nulling the educational benefits they could provide (Kerawalla and Crook, 2005; Van Eck, 2006; Charsky, 2010; Floyd and Portnow,

[Figure]

**Fig. 1.** reviewers comments on PDF

---

## Author Comment (AC1) · 18 Oct 2020

Dear Ian, we would both firstly like to say thank you for reviewing our manuscript and for the positive words in response to it.

We also welcome all of your suggestions to enhance our paper. Obviously the topic of video game learning is very expansive, particularly with all the different styles of gaming out there each providing a different learning experience. Therefore, we will not be able to cover anything in a single paper. However, the bases that you provided, the

age barriers and MMORPG communities really got us thinking and wondering how we missed that as a point of discussion.

I strongly believe that we can take your review on aboard to create the best manuscript possible, and continue the advise for our follow up paper.

---

## Author Comment (AC2) · 18 Oct 2020

Dear Jamie, we would firstly like to say thank you for reviewing our manuscript and we are very glad you enjoyed it, particularly the video supplement. We were unsure of how it would be received but based on your comments we are more than happy to try and add more in of other hazards. Unfortunately, we have lost the raw video of a layer sequence, however, we still have some edited gifs of the sequence that maybe of sufficient quality.

[Figure]

The summary table matrix is also a very good suggestion, which I can see will help improve our conclusion for readers. I believe two tables, one containing volcanic aesthetics and the other containing volcanic hazards, would best cover all the features that we found while reviewing the video games.

Hopefully we will be able to take on all of your suggests to create the best manuscript possible.

---

## Short Comment (SC1) · 21 Oct 2020

Dear authors and editor(s),

I am submitting this unsolicited review as I came upon this pre-print on twitter and immediately gave it a read. The topic is very exciting, and I think this paper will make an important contribution to the literature. I had some thoughts while reading the paper and wanted to suggest some ideas in case the authors feel they will improve the manuscript.

1. More substantive data analysis. I realize that constructing a quantitative analysis is difficult in this context, but I think some more data could be gathered and used as a statistical showcase of the state of the representation of volcanoes in COTS. For example, without needing to do a full review of more games, the authors could comb through other COTS and create a database of games including their title, release date, genre/style, and importantly, which volcanic features are represented. Perhaps an accuracy score (say, 1-5 with 5 being most accurate, as deemed so by the authors) could also be tabulated for each feature, but this may be too subjective. In any event, I would have loved to have seen, for example, a histogram of the number of occurrences of volcanic edifices, lava flows (active), lava flows (lithified), ash fall, ash deposits, PDCs, lahars, etc. Even if this doesn't reflect the actual abundances of these features in all games, it would still illustrate the abundances of these features within the games studied. It would be simple to add more games to this list that would not require full reviews, such as Star Wars Battlefront, No Man's Sky, DOOM, even as far back as Super Mario (with pits of lava contained within castles). This effort could probably also be crowdsourced!

Also, what portion of the total number of COTS is this representing? Surely the authors cannot play every game, but is there a way to estimate the proportion of volcano-bearing games to the total number of games? How about based on genre? (e.g., 15% of action adventure games contain volcanic features, whereas only 2% of racing games).

These are just example ideas, my point being that I think more of this type of analysis could help the paper to contextualize the study within the broader subject of COTS.

2. Direct comparison of video game features to real world features. The authors do a nice job of describing some of the accurate features showcased in COTS, but I think this could be explored slightly further. For example, I would suggest a figure with real pahoehoe lava flows compared to in-game pahoehoe flows. The non-volcanologist reader of this work I think would benefit from seeing how these in-game features compare to real life.

3. Limitations and advocacy for realism in games The authors allude several times to the limitations of making volcanic features realistic in video games. It got me thinking about the perspective of game developers. Is there a conversation to be had here on how and why decisions regarding the realism of volcanic features in COTS are made from a development perspective? While this is perhaps beyond the scope of the article. But, since it seems the authors have given some thought to the give and take of creating realistic and engaging video games, I'm curious to know where the authors stand. Do the authors advocate for more realism? The same amount of realism? Are there specific pitfalls developers should avoid? Which inaccuracies are the most harmful, and which are perhaps fine to exaggerate for story purposes?

Overall, I think this paper is a very nice contribution to the literature and congratulate the authors on a nicely put together manuscript. Thanks for the opportunity to chime in, despite the fact that no one asked me to! I hope the comments are well received.

Very best, Kayla Iacovino

---

## Short Comment (SC2) · 23 Oct 2020

Dear Kayla,

Many thanks for your comments and we are glad you enjoyed the paper. Ed and I have discussed what we shall do in regards to your suggestions:

1. We feel that your suggestion for a more substantive data analysis, whilst it would add another dimension to this paper, is beyond the scope of this study. However, we will take this into consideration for future studies.

[Figure]

- The histogram suggestion of the number of occurrences of various volcanic features is a similar suggestion to reviewer 2's idea of a summary table. We will therefore, take this on board and add this to the paper.

- On the suggestion of scoring, we believe this would be subjective. We have done something similar on our blog posts and even then, it has been purely based on our own personal opinions. This would not be appropriate for a peer reviewed paper.

- We do like the idea of crowdsourcing the help to generate a more comprehensive list of COTS games that feature volcanism. We did try to do this before this paper was written however, we did not get any engagement. Now that we do have this paper, we will attempt to generate interest in making this a more collaborative effort and start this process on a blog post, before expanding into a peer reviewed paper.

2. This is an excellent suggestion that we had not considered previously. We shall take this on board and provide figures with real world volcanic features to visually compare to the video game counterparts.

3. Whilst this is an interesting and valid suggestion, this would perhaps be better suited for a separate paper with the findings linked to this paper. We have previously considered speaking with game developers and ask such questions that you have suggested for future work.

- Ed and I are unsure on our stance on whether games should be more realistic or not. This would depend on the developer's reasonings and introduce a degree of subjectivity.

I hope this is a satisfactory response to your comments, please do let us know if you need further explanations.

Kind regards,

Jazmin Scarlett

---

## Editor Comment (EC1) · Steven Rogers (Editor) · 3 Nov 2020

Declaration: I am the handling editor for this manuscript, but this comment is as an interested individual, please treat it as such (i.e. these are not conditions you must meet, just thoughts that are hopefully of use)!

Hello Edward and Jazmin - I really enjoyed reading this submission and think this type of study, linking geology to 'everyday society', is a really important avenue geoscientists need to engage with.

[Figure]

Other comments have included some nice suggestions about the framing of the paper - my comments/thoughts are more pedagogy focused:

Section 1.4 This section does a nice job of introducing tangential and incidental learning, I think you could consider expanding the section a little to include some key references, examples and possibly case studies - setting up these concepts here would allow a more detailed discussion (I'm not saying you don't do this, section 4.2 starts to explore this) of the learning happening whilst playing COTS - and really set this paper up as a springboard to your planned second paper (which sounds like an interesting read!). There are several papers available discussing tangential/incidental learning in both educational games and COTS.

Section 4.3 This section could consider if this matters in the geosciences - does it matter if people pick up erroneous facts from playing games (can these facts be easily "corrected" in a formal educational setting? Are there any studies on this?) Could it be more/as important that people are being exposed to geoscience through COTS, even if it is erroneous?

Hopefully some useful suggestions?

Cheers,

Steve Rogers

---

## Author Comment (AC3) · 11 Nov 2020

Hi Steven,

As always, it is great to hear such praise for our work and to hear that people are not only interested in the topic, but believe that others should be too.

For your comments, we believe that your suggestions for Section 1.4 are extremely valid and could definitely help to strengthen not only this section but the discussions later on as well.

[Figure]

For the Section 4.3 suggestions, we believe this is just slightly out of the scope of this paper, but will definitely be one of the main focuses for our next paper. In that one we hope to discover what people truly learn from playing COTS, and in turn, whether it matters. Therefore, the questions you proposed to us will be much more useful for that paper.

Thank you very much for your feedback.

Cheers, Ed

---

## Short Comment (SC3) · 21 Nov 2020

The manuscript documents the appearance of volcanoes in commercial off-the-shelf video games and considers how the accuracy of their portrayal may impact the learning and perception, about volcanoes, of those playing the games. This is a very well written and interesting manuscript which raises important questions and considerations. I feel that it should be published with some minor corrections as suggested below.

The analysis seems concentrated predominantly on the styles of Role-Playing Games

[Figure]

(RPG) and Action-Adventure which may necessarily preclude the full suite of volcano representation across other game types. Perhaps I missed it but I could not see a justification of why these types of games were selected as oppose to others. Or, also, perhaps I am doing a disservice to the spectrum of game styles chosen as I simply don't know them all. From my, limited experience, of commercial gaming I have noted volcanoes in several strategy and simulation games.

A recent example that could be explored is the use of natural disasters, such as volcanoes, in the simulation game by Limbic Entertainment - 'Tropico 6' which sees players act as the President (or dictator – depending on the style of governance chosen) of a series of small fictional Caribbean type islands. In fact, all of the Tropico franchise games contain volcanoes in the various islands/levels but, in my opinion, tropico 6 provides the most realistic representations. In the level called 'Penultimo of the Carribean' the player constructs their society of an archipelago of islands of which is clearly modelled on 'White island' volcano in New Zealand' (Image attached – no. 1). The island exhibits steaming fumaroles and murky coloured water offshore. The level called 'Pirate King' hosts what appears to be a volcano with a persistent lava lake (image attached – no. 2). The level 'Acts of god' is based entirely around the occurrence of natural disasters with a 'Krakatau' style volcano called 'Mount Kraken' in the centre of a series of playable islands (image attached – no. 3,4 and 5). In this 'level' the player must manage their island whilst consistently overcoming the destruction resulting from volcanic eruptions. This adds another dimension to the understanding or interaction of volcanoes in games being the socio-economic situation that contributes to disasters. It is an interesting parallel that depending on how well the player governs their island will ultimately dictate how disastrous the volcanic consequences are. I think this is apt since it volcanic risk is comprised both of the hazard and the vulnerability of society. The challenge in the game is then, at least partly, to build resilient communities.

Another important commercial game in the action-adventure genre that featured volcanoes, which was not discussed, was one of the first releases on the Nintendo Wii

console, namely, Disaster: Day of Crisis. It was a while since I played the game but from memory the first level and opening scenes were set on the slopes of an erupting volcano.

Of course, it is very likely that there are many more games that contain volcanoes and I wouldn't expect the authors to discuss or play them all as there would be a significant cost involved in purchasing them all presumably, but I think some justification of why those selected were chosen and others not, and perhaps a discussion of the historical importance of volcanoes in games could be valuable. When for example did volcanoes first appear in games and has their illustration or action changed with time, and how?

Minor comment:

Paragraph 335

Please give the full names of the societies along with the abbreviations, i.e., American Geophysical Union (AGU).

Regards, John Browning
* * *
[Figure]

[Figure]

**Fig. 1.**

[Figure]

**Fig. 2.**

[Figure]

**Fig. 3.**

[Figure]

**Fig. 4.**

[Figure]

**Fig. 5.**

---

## Author Comment (AC4) · 24 Nov 2020

Dear John, I would first of all like to say thank you for taking time to read our manuscript. We are very glad that you enjoyed it so much.

To address your comments, the games we covered were chosen purely for their popularity, either as individual games, part of well-known franchises, or came highly recommended by friends during the early stages of the project. It just happens to be the case that a majority of the video game market at the moment is dominated by RPG and

[Figure]

Action-Adventure games. Even looking at online forums giving lists of 'Top 10 video game volcanoes' they are mostly populated with these types of games. We do briefly mention our choice of popular games in Section 2, but I am happy to expand on it to make it clearer.

One game that we did cover, From Dust, is classed as a strategy/ 'god' game, where you can control the elements to prevent natural disasters such as volcanic eruptions and tsunamis from destroying native civilisations. This was not previously mentioned in the manuscript purely due to other games making better examples of topics we cover. However, this is now going to be included in a summary table of all the games we have reviewed.

We did also think about Mario Kart (a racing game), which has several volcanic race tracks including Rumble Mountain and Bowser's Castle. Unfortunately, neither of us had a copy to hand and even though the latest edition on the Nintendo Switch was released a few years ago now, it has yet to significantly decrease in price to consider purchasing just to review.

I must admit I have not heard of the two examples that you provided. However, your in-depth description and images of Tropico 6 have definitely intrigued me. Perhaps with much more time, we will be able to expand our work to form a database of video games and cover a broader range of genres.

I believe your suggestion of a discussion of the historical importance of volcanoes in video games and their development through time is just outside the scope of this particular paper. However, we have been discussing with others to look further into the development process of commercial games as part of another manuscript, and I feel this would fit in perfectly with your suggestions.

Finally, full names of the societies will of course be added along with the abbreviations. Thank you for pointing that one out to us.

I hope we have been able to answer your questions and will be able to input your suggestions appropriately.

Kind regards, Edward McGowan
* * *

---

## Author Response (AR2)

**Point-by-point Reply to Comments**

**Responses to Steve Rogers' initial Editor Decision:**

Firstly, the opening statement is very positive and does feel encouraging.

- The request for technical corrections is understandable and valid. We did not consider the necessity to reference the games we used. However, as the Editor has pointed out, they are the no different to using other forms of media.
- The Editor's use of examples for a referencing style made the editing very easy to undertake and left little to question in his intents for us.
- The final statements are once again very encouraging, and we too look forward to the review and discussion.

**All Relevant Changes**

The following video game references were added to the Reference List:
- Assassin's Creed: Odyssey (Standard Edition), 2018.
- Crash Bandicoot N. Sane (Standard Edition), 2018.
- LEGO DC Supervillians (Standard Edition), 2018.
- LEGO Marvel Super Heroes 2 (Standard Edition), 2018.
- Monster Hunter Generations Ultimate (Standard Edition), 2018.
- Pokémon Emerald (Standard Edition), 2005.
- Sea of Thieves (Standard Edition), 2018.
- Spyro: The Reignited Trilogy (Standard Edition), 2018.
- Subnautica (Standard Edition), 2018.
- The Elder Scrolls: Skyrim (Special Edition), 2016.
- The Legend of Zelda: Twilight Princess (HD Edition), 2006.
- The Legend of Zelda: Breath of the Wild (Standard Edition), 2017.
- The Shadow of Tomb Raider (Standard Edition), 2018.

The above video games were also fully referenced throughout the main text, with dates added. Supplement Video DOI added to the Reference List.
Author Contribution and Competing Interests sections added.

**Responses to Steve Rogers' final Editor Decision:**

- As pointed out by Steve, we have agreed with all of the comments made by the Reviewers and scientific community who read our paper that is to be required for the final version of the manuscript.
- We strongly agree with the necessity for a summary table and real-world/authentic images of volcanic features. They have helped to add more relevance to our research and the wider community.
- The additional information outlining the two forms of learning we mention and our reasons for picking the game we did has also strengthened the scientific content of this revised manuscript.
- Finally, we have taken on a few other suggestions that we made during the preprint scientific discussion stage, including two more supplementary videos and bar charts.

**All Relevant Changes**

- Section 1.2. was added as per the suggestion of RC2.
- Section 1.4 & 1.5 renumbered due to the addition of Section 1.2.

- Section 1.5 contains more examples and references related to how COTS games can educate through tangential and incidental learning (EC1).
- Section 2.1 added to explain our choice of COTS video games reviewed for this paper (RC1 & SC3).
- Fig 01, 02, 07 & 08 contains additional images of authentic volcanic features (SC1).
- Video Supplement 01 & 02 included as per the suggestion of RC2. Video Supplement 03 (originally 01) has had its number changes to reflect its occurrence in the paper.
- Fig 10, a bar chart of features and hazards documented in all the video games added as per the suggestion of SC1.
- Section 4.1 includes addition text to talk about results seen in Fig 10.
- Section 4.4 includes extensive text added to talk about how COTS games could be implemented within educational and outreach settings (RC1 & RC2).
- Full names of geological societies included as per request of SC3.
- Section 4.5 expands fully on the considerations needed to implement COTS games into educational settings (RC1).
- Tables 01 & 02 are summary tables of all the aesthetic features and hazards documented within each video game reviewed for this paper (RC2).
- Additional references included.

**Responses to Steve Rogers' final Technical Corrections:**

All comments were clearly worded, and corrections were easy to locate and fix.

**All Relevant Changes**
- Typo removed from Pg 2 L 69.
- Terminology corrects on Pg 3 L 08.
- Individual titles added to Fig 10a and 10b.
- Sentence reworded on Pg 17 L 06.
- Sentence reworded on Pg 17 L 08.
- Additional references to Leeds and Imperial's work on virtual mapping systems added to Pg 18 L 26.
- Italics added and removed to the correct part of the game reference on Pg 19 L 54 & 55.

[revised manuscript text omitted]